# Characterization of Anaerobic Rumen Fungal Community Composition in Yak, Tibetan Sheep and Small Tail Han Sheep Grazing on the Qinghai-Tibetan Plateau

**DOI:** 10.3390/ani10010144

**Published:** 2020-01-16

**Authors:** Wei Guo, Weiwei Wang, Sisi Bi, Ruijun Long, Farman Ullah, Muhammad Shafiq, Mi Zhou, Ying Zhang

**Affiliations:** 1State Key Laboratory of Grassland Agro-ecosystems, College of Pastoral Agriculture Science and Technology, Lanzhou University, Lanzhou 730020, China; guow16@lzu.edu.cn (W.G.); wangww14@lzu.edu.cn (W.W.); 2School of Life Sciences, Lanzhou University, Lanzhou 730020, China; biss16@lzu.edu.cn (S.B.); longrj@lzu.edu.cn (R.L.); 3Department of Animal Breeding and Genetics, Lasbela University of Agriculture Water and Marine Sciences, Uthal 90150, Pakistan; aup.veterinarian@gmail.com; 4College of Veterinary Medicine, Nanjing Agricultural University, Nanjing 210095, China; drshafiqnjau@yahoo.com; 5Department of Agricultural, Food and Nutritional Science, University of Alberta, Edmonton, AB T6G 2P5, Canada; 6School of Public Health, Lanzhou University, Lanzhou 730020, China

**Keywords:** anaerobic rumen fungal community, yak, Tibetan sheep, Small Tail Han sheep, Qinghai-Tibetan plateau, dynamic association

## Abstract

**Simple Summary:**

Anaerobic rumen fungi play a vital role in fiber degradation. The objective of this study was to compare the anaerobic rumen fungal communities of full grazing ruminants in the Qinghai-Tibetan Plateau. Our results showed that the anaerobic rumen fungal community was affected by host species and the dynamic associations of them were host specific. This is the first study exploring the anaerobic rumen fungi in the full-grazing ruminants, which could lay a solid foundation to really identify fiber degradation fungal taxa using culture-dependent techniques in the future.

**Abstract:**

The anaerobic rumen fungal community play a critical role in fibrous material degradation. However, there is a lack of data describing the composition of anaerobic rumen fungal community of full grazing ruminants in the Qinghai-Tibetan Plateau. For this reason, we employed the next-generation sequencing technique to elucidate the rumen fungal structure composition and evaluate the effects of host species on fungal communities. Community comparisons (Bray–Curtis index) between yak and Tibetan sheep revealed that the rumen fungal community was affected by host species (*p* < 0.05). The alpha diversity indices in the yak were significantly higher than in the Tibetan sheep and Small Tail Han sheep. Neocallimastigomycota was predominant regardless of host species. Within this phylum, unidentified genus of Neocallimastigaceae was the most dominant in all samples, followed by *Piromyces* and *Orpinomyces*. Moreover, the shared and unique OTUs in the rumen were identified and most of them belonged to the *Orpinomyces*. Co-occurrence network analysis identified that each animal species had their own keystone species and most of them were non-dominant flora. Our data indicate that host breeds override living environment as the key factor that determines fungal community in the rumen of grazing ruminants in the Qinghai-Tibetan Plateau.

## 1. Introduction 

Anaerobic fungi were first isolated from the ruminants in the 1970s [1] and at least 50 different herbivorous animal species host anaerobic fungi in their gastrointestinal tract [2]. Currently, nine cultivated anaerobic fungal genera have been identified with many other uncultured taxa known to exist [3,4], and all of them belong to the order Neocallimastigales within the phylum Neocallimastigomycota [5]. Only 29 species of anaerobic fungi have been identified using culture-dependent techniques [4]. Anaerobic fungi, despite their low percentage (10–20%) of ruminal microbes based on rRNA transcript abundance [6], play a vital role in fiber degradation, primarily due to their efficiency and their producing of an extensive set of polysaccharide-degrading enzymes for plant materials degradation [7,8]. Moreover, rumen fungi have amylolytic and proteolytic activity [9]. Their rhizoidal system can physically penetrate plant cell walls and secrete a wide variety of extracellular polysaccharide-degrading enzymes and phenolic acid esterases to degrade lignocellulose [10], consequently, releasing a great amount of H_2_ that favor the archaeal community [11]. Thus, anaerobic fungal activity may have the capability to shape the composition of bacterial and archaeal communities in the mature rumen, as a result, affecting fiber utilization efficiency and methanogenesis [12,13].

Early study into anaerobic rumen fungi of goat has revealed that the concentration and diversity of them are associated with a fiber-rich diet [14]. Furthermore, many studies have identified that anaerobic rumen fungi not only decreased the voluntary feed consumption of sheep when the anaerobic fungi were removed from the rumen [15], but also affected the dry matter digestibility [16], feed efficiency and milk production [9]. Therefore, understanding of anaerobic fungal communities is very important to utilize fibrous material, aiming to improve feed efficiency.

Yak and Tibetan sheep are indigenous animals in the Qinghai-Tibetan Plateau that is characterized by cold weather conditions and mountainous terrain with altitudes ranging from 4000 to 5500 m, which provide milk, meat, fuel (yak dung), and wool for local herdsmen [17]. Small Tail Han sheep, which originated from low land [18], later adapted to the grazing lifestyle on the Qinghai-Tibetan Plateau and became an indispensable species to the local life and economy [19]. Although a few studies have reported the rumen prokaryotic community from grazing yak, Tibetan sheep and Small Tail Han sheep [19,20,21,22,23], few studies have considered the rumen eukaryotic community composition of these three ruminant species, especially grazing under the same natural pasture. Thus, our prime aim of this study is to characterize the rumen fungal community of mainly grazing ruminant species (Yak, Tibetan sheep and Small Tail Han sheep) in the Qinghai-Tibetan Plateau, so as to identify the ruminal fungal strains that are prevalent in the rumen which may be associated with fiber degradation.

## 2. Materials and Methods 

### 2.1. Animals and Sampling

The experimental procedures were conducted in accordance with the Animal Ethics Committee of the Chinese Academy of Lanzhou University (permit number: SCXK Gan 20140215). In summary, rumen samples were obtained from 18 full-grazing ruminants on the Qinghai-Tibetan Plateau (September, 2017), including 6 castrated yaks, 6 castrated Tibetan sheep and 6 castrated Small Tail Han sheep, with an average body weight of 200 ± 20 kg, 54 ± 2.7 kg and 46 ± 4.2 kg, respectively (Ave ± SD). The experimental animals were grazed all year round at Wushaoling in the Qinghai-Tibetan autonomous county of Tianzhu, Gansu Province (37°12.4′ N, 102°51.7′ E, 3154 m a.s.l). The local temperature ranges from −8 °C to 4 °C annually. The rumen contents were collected via rumen cannula before the morning grazing followed by filtering the large-sized particle materials using sterilized cheesecloth, and then the samples were stored at −80 °C prior to DNA extraction. 

### 2.2. DNA Extraction

The total genomic DNA was extracted from the rumen fluid samples using the PowerSoil DNA Isolation Kit (Mo Bio Laboratories, Inc., Carlsbad, CA, USA) according to manufacturer’s instructions. After that, the Qubit^®^ 2.0 Fluorometer (Life Technologies, Eugene, OR, USA) was used for DNA quantification and then stored at −20 °C for further microbial analysis.

### 2.3. Amplification Sequencing and Statistical Analysis 

For analyzing rumen fungal profile, primers MN100F (TCCTACCCTTTGTGAATTTG) and MNGM2 (CTGCGTTCTTCATCGTTGCG) targeting internal transcribed spacer region of anaerobic fungi [24] and the PCR amplification were performed according to the conditions described by previous study [25]. After quality control and purification, the qualified PCR products with bright main strip were chosen for sequencing on an Illumina HiSeq2500 platform to generate 250 bp paired-end reads. The paired-end reads were demultiplexed according to the corresponding unique barcodes followed by merging the reads using FLASH (V1.2.7, http://ccb.jhu.edu/software/FLASH/) [26]. The quality control was performed according to the QIIME (V1.9.1, http://qiime.org/index.html) pipeline [27]. Then, the chimeric sequences were removed using USEARCH software based on the UCHIME algorithm [28]. OTUs were picked using open_reference OTU picking method and the UNITE dynamic ITS database (v01.12.2017) [29] were used as reference taxonomy [30]. The singleton OTUs were filtered prior to facilitate further analysis. The following data analysis was performed within the QIIME pipeline software. In brief, the α-diversity indices (within sample comparison), including Shannon (diversity), Chao1 (richness), Good’s coverage and the estimated OTUs, were analyzed using the default parameters in the QIIME software. As the fungal ITS region is not amenable to phylogenetic analysis [31], the principal component analysis based on Bray−Curtis metric was calculated to measure the dissimilarity of microbial communities among animal groups and were plotted using package ‘ade4’ of software R (v3.2.3) [32]. Furthermore, the shared and unique OTUs among different animal species (prevalent in at least three samples, 3 out of 6) were calculated and visualized using the VennDiagram package in R software. To reduce rare taxa in the data set, only the taxa with relative abundances more than 0.01% and occurring in at least 50% of all samples (3 out of 6) within each animal species were included in the co-occurrence network analysis. Specifically, the Spearman correlation matrix was constructed by WGCNA package [33]. To better reflect the entire microbial network, the correlations with the R-corr absolute values more than 0.1 were all presented in the network figures without filtering by *p* values, and the corrections with statistical significance were highlighted in the figures with asterisks. The topological features were calculated with igraph package [34]. The Nodes with the highest betweenness centrality scores were considered as keystone species in the network [35]. The network image was generated using Cytoscape (https://cytoscape.org/). All the data were presented with Ave ± SD unless indicated otherwise, and statistically significant differences were declared at *p* < 0.05, the Wilcoxon test was used to determine the significant level between two animal groups, while Kruskal−Wallis test was used to calculate the significant level among animal groups, and the *p* value was adjusted using Benjamini–Hochberg method. All the sequence data were subjected to the Sequence Read Archive (SRA) of the NCBI under BioProject ID PRJNA573900. 

## 3. Results

### 3.1. Sequencing Profile

After quality control and chimera removal, a total of 1,529,753 high-quality fungal sequences were obtained, which were assigned to 640 non-singleton OTUs. In particular, 81,711 (±3632) effective reads were obtained in the yak that yielded 210 (±27) OTUs, while 83,826 (±5103) and 89,421 (±7070) sequences were identified in the Tibetan sheep and Small Tail Han sheep, respectively, which separately clustered into 162 (±25) and 165 (±41) OTUs. The Good’s coverage [36] ranged from 99.90% to 99.94%, indicating the sequencing depth sufficiently covered the diversity of fungal communities in all samples. A summary of sequence count and OTU number per sample was shown in Appendix A.

### 3.2. Effects of Animal Species on Rumen Fungal Community and Diversity 

Host species-related differences were apparent in the fungal community when visualized by Principal component analysis (PCA) based on Bray−Curtis distance metrics (Figure 1A). By comparing yak and Tibetan sheep, the rumen fungal community was significantly different (Bonferroni-corrected *p* < 0.01). However, no such difference was observed between yak and Small Tail Han sheep. Furthermore, individual variations in the yak and Tibetan sheep were lower than in the Small Tail Han sheep, given the closer distance between samples in the yak and Tibetan sheep. As shown in Figure 1B, 65 OTUs were shared by three animal species. Of these, 17 belonged to genus *Orpinomyces* and 22 OTUs were classified into unidentified Neocallimastigaceae; the other belonged to genera *Anaeromyces, Caecomyces, Cyllamyces, Neocallimastix* and unidentified taxa. The number of OTUs (n = 31) shared between yak and Small Tail Han sheep was close to that (n = 30) shared between Tibetan sheep and Small Tail Han sheep. In addition, 98 OTUs were exclusively identified in the yak, and among them, 38 OTUs belonged to genus *Orpinomyces*. There were 43 OTUs exclusively detected in the Tibetan sheep, 16 of them belonged to genus *Orpinomyces*. As to Small Tail Han sheep, only 12 OTUs were exclusively identified and 5 of them were assigned to genus *Orpinomyces* (Figure 1B). To sort out the effects of animal species on alpha diversity, including Shannon index (diversity), Chao1 (richness), and Observed_species (estimated OTUs) were measured. The Shannon diversity in the yak was observed significantly higher than in the Tibetan sheep and Small Tail Han sheep (Benjamini–Hochberg *p* < 0.05, Figure 2). The richness in the yak was significantly higher than in the Tibetan sheep and Small Tail Han sheep (Benjamini–Hochberg *p* < 0.05, Figure 2). Remarkably higher estimated number of OTUs were found in the yak compared to that of Tibetan sheep and Small Tail Han sheep (Benjamini–Hochberg *p* < 0.05, Figure 2).

### 3.3. Comparison of Anaerobic Rumen Fungal Composition in Three Ruminant Animal Species

A total of 2 fungal phyla were identified, Neocallimastigomycota and Ascomycota. Among them, the Neocallimastigomycota was the most dominant phylum, accounting for 92.86% (± 0.094) of total sequences (Figure 3). The relative abundance of Neocallimastigomycota in the Tibetan sheep (97.44 ± 0.007%) and Small Tail Han sheep (90.65 ± 0.158%) was higher compared to that of yak (90.49 ± 0.035%, Appendix A). The relative abundance of unidentified phylum in the yak (5.91 ± 0.02%) was significantly higher than in the Tibetan sheep (1.08 ± 0.007%) and Small Tail Han sheep (0.76 ± 0.004%, Benjamini−Hochberg *p* < 0.01, Appendix A).

Our taxonomic composition analysis of rumen fungi at the genus or higher level identified 11 taxa. Unidentified Neocallimastigaceae (53.08 ± 0.234%) was predominant, regardless of animal species, and its relative abundance remained stable among different animal species but varied among individuals (Figure 4). The relative abundance of genus *Piromyces* in the Tibetan sheep (31.54 ± 0.237%) was significantly (Benjamini–Hochberg *p* < 0.05) higher compared to yak (2.58 ± 0.021%, Appendix A), while there was no significant difference between Tibetan sheep and Small Tail Han sheep (17.77 ± 0.19%). The frequency of genus *Orpinomyces* was higher in the yak (24.67 ± 0.045%) compared with Tibetan sheep (13.09 ± 0.102%) and Small Tail Han sheep (13.37 ± 0.094%, Appendix A). Other rare genera (relative abundance < 5%) were also identified in the present study such as genera *Anaeromyces*, *Caecomyces*, *Cyllamyces*, *Neocallimastix* and *Oontomyces* (Figure 4). Among them, the incidence of *Neocallimastix* was significantly higher in samples from Tibetan sheep (2.47 ± 0.014%) compared with yak (0.05 ± 0.001%, Appendix A, Benjamini–Hochberg *p* < 0.05).

### 3.4. Co-Occurrence Network Analysis 

The fungal taxa inter-interactions within each animal species were calculated based on the Spearman’s correlation and the importance of each node in the network was ranked according to betweenness centrality and other topological features. In the yak, unidentified genus of Neocallimastigaceae and unidentified fungi were the keystone species, presenting the highest betweenness centrality (Appendix A), indicating the relevance of these two nodes as capable of holding together communicating nodes, and they had strong negative correlation with each other. In the network, the highest positive correlation was identified between genus *Cyllamyces* and unidentified Neocallimastigaceae (*cor* = 0.89, *p* = 0.019), while the predominant negative correlation was detected between genera *Anaeromyces* and *Neocallimastix* (*cor* = –0.94, *p* = 0.005, Figure 5A). As for Tibetan sheep, *Anaeromyces*, *Cyllamyces*, *Piromyces* and unidentified genus of Neocallimastigaceae were considered as keystone species based on the betweenness centrality (Appendix A). The *Caecomyces* and *Neocallimastix* had the predominant positive correlation (*cor* = 1, *p* = 0), whereas *Piromyces* and unidentified genus of Neocallimastigaceae showed the strongest negative correlation (*cor* = −0.94, *p* = 0.005, Figure 5B). In case of Small Tail Han sheep, no Blast hit and *Orpinomyces* were treated as keystone species based on the betweenness centrality (Appendix A). The positive correlation between *Caecomyces* and *Piromyces* was much stronger than others (*cor* = 0.94, *p* = 0.005), whereas no Blast hit and unidentified Neocallimastigaceae had the predominant negative correlation (*cor* = −0.94, *p* = 0.005, Figure 5C).

## 4. Discussion

The objective of this study aimed at comparing the anaerobic rumen fungal community as well as the dynamic associative patterns of them in the dominant ruminant species in the Qinghai-Tibetan Plateau. Findings from the present study profile the anaerobic rumen fungi of dominant grazing ruminant species (Yak, Tibetan sheep and Small Tail Han sheep) in the Qinghai-Tibetan Plateau, which gains a better understanding of taxonomic composition and the dynamic associations of rumen fungi within each ruminant species. Owing to the sampling method we have applied, the findings from the current study were restricted to the planktonic community rather than the entire rumen community. However, in industrial practice, the tubing method is more frequently used for large ruminants grazing on pasture to collect rumen samples, through which rumen liquid will be collected and examined. Our results are, therefore, similar to the nature of those samples and more comparable with those studies.

Although the experimental animals grazed on the same natural pasture, based on the PCA analysis, we found that there were clear, distinct differences in the composition of anaerobic rumen fungi between yak and these two sheep species, while no such separation was found between Tibetan sheep and Small Tail Han sheep. Similar findings were observed in the previous study on other ruminant species such as wether sheep, mature non-lactating dairy and mature castrated red deer [13], where the fungal community varied by ruminant species. Naturally, the Tibetan sheep and Small Tail Han sheep graze together; this may make their fungal communities more similar to each other given the fungi are derived from the environment. These results suggest that anaerobic fungal communities do not randomly assemble in the rumen, but that different species occupy distinct environmental niches influenced by the host animal.

The yak, being the largest animal in the current study, had the largest number of unique fungal taxa, followed by Tibetan sheep and Small Tail Han sheep. Similar results were reported on rumen bacteria in the same habitat for wild boreal cervids [37], where moose had the largest number of unique bacteria; the second largest animal, being deer, presented the second largest number of unique bacteria and the smallest species, roe deer, had the lowest number of unique species. The underlying explanation for these differences could be the host physiology that treated, as a contributing factor, for the sharing of the microbiota between the two largest animals [37]. However, the specific mechanism of this phenomenon needs to be verified by further experimentation.

Alpha diversity metrics (Shannon, Chao1 and Observed_species) displayed a higher diversity in yak compared to Tibetan sheep and Small Tail Han sheep. Our previous study on rumen bacteria community using the same animals [19] as well as another study on grazing yaks and Tibetan sheep [38] found the similar result on alpha diversity indices, indicating that the animal body size or diet may have the similar effect on prokaryotic and eukaryotic microorganisms in rumen. Further studies could be done on rumen archaea and protozoa to warrant this hypothesis.

The prevalent microorganisms included fungi, belonging to the phylum Neocallimastigomycota, in all animals in the present study. Previous study on yak from intensive farming [39] found that the most dominant phylum was Ascomycota in the rumen. This discrepancy may be due to feeding system (grazing vs. captive) or diet (natural grass vs. concentration). In addition, reports on dairy cows and Chinese Holstein dairy cows also found that phylum Ascomycota was predominant in the rumen [30,40]. This result may be due to the host selective effect on its symbiotic fungi. Moreover, anaerobic fungi (Neocallimastigomycetes) play an important role in converting lignin-rich plant biomass into sugars in the rumen of animals [4,41,42,43], representing a very promising enzyme resource to contribute to the conversion of plant biomass into biofuels, which indicates that the grazing ruminants in the Qinghai-Tibetan Plateau may have more potential ability to utilize the recalcitrant material. Furthermore, we found that more proportion of unidentified taxa existed in the yak than in the Tibetan sheep and Small Tail Han sheep, which indicates that yak have more novel taxa which could be explored in the future [44]. More studies based on the culture method should be performed to really identify the useful strain from unidentified Neocallimastigaceae.

At the genus or higher level, we found the unidentified genus of Neocallimastigaceae was the most predominant regardless of animal species, which is consistent with previous study on dairy cows [30]. Recent studies on dairy cows have found that *Caecomyces* was the second most dominant genus [11,45], while in the present study, *Piromyces* was the second prevalent genus in the rumen, irrespective of animal species, but the relative abundance of genus *Caecomyces* was very low (0.76%). Previous study has reported that the genus *Piromyces* can produce cellulolytic and hemicellulolytic enzymes [46,47], while members (*Caecomyces communis*) belonging to genus *Caecomyces* had glycoside and polysaccharide hydrolase activity. This result indicates that such fibrolytic ruminal fungal strains are prevalent in grazing ruminants such as grazing yaks, Tibetan sheep and Small Tail Han sheep, and could be exploited as microbial feed additives to improved fiber utilization in ruminants from intensive farming. In addition, we found that the relative abundance of genera *Piromyces* and *Neocallimastix* were significant higher in Tibetan sheep in comparison to yak, suggesting that the Tibetan sheep have the relatively high ability to utilize recalcitrant material (cellulose and hemicellulose) considering these two genera are capable of producing cellulolytic and hemicellulolytic enzymes [46]. Additionally, many previous studies have discovered genus *Neocallimastix* was dominant in many ruminant species, such as American bison, sheep and llama [5,24,47], which was less abundant in the current study. This discrepancy in the fungal populations observed are inevitable as feeds and breeds of experimental animals affect population dynamics. It should be noted though, that the primers used in the current study have been used in the previous study [24] and in one of our previous studies [25], where sheep, cattle and deer were examined. This primer pair has successfully generated amplicons from all samples collected from the current study and was therefore being adopted. However, it should be noted that this primer set may have generated biased results where Neocallimastigomycota is not well represented [3]. Therefore, in the future it is necessary to either choose additional primers or to examine the metagenome to obtain the more complete rumen fungal profiles and to assess the functional potentials of the anaerobic rumen fungi.

Finally, the inter-interactions of rumen fungi of each animal species were explored. As expected, each animal species had unique keystone species and the connectivity of network was different from each other. Specifically, genera *Caecomyces* and *Cyllamyces* had positive correlation in the Tibetan sheep network, which is similar to previous study on dairy cows [30], that these two taxa co-occurred with each other. However, in the network of yak and Small Tail Han sheep, these two genera had negative correlation. These results suggest that the dynamic association of rumen fungi may be influenced by animal species. Moreover, the relationship between *Cyllamyces* and *Piromyces* was negative across animal species, which is inconsistent with the report on Nordic Red dairy cows [48] where these two taxa had strong positive correlation, indicating that not only the animal species contributes to modulate the interaction of rumen taxa, but also the living environment play an important role in defining the dynamic associative patterns. In addition, most of the keystone species in the network were less abundant except that in the yak, and the stronger associations were always found between the less abundant taxa. These results indicate that non-dominant taxa may play an important role in the rumen ecosystem.

## 5. Conclusions

In summary, the present findings showed distinct shifts in the rumen fungal communities in yak, Tibetan sheep and Small Tail Han sheep. We found that fibrous ruminal strains are prevalent in these three grazing ruminant species. In addition, the shared and unique OTUs were identified in the rumen samples of these three ruminant animals. The co-occurrence network analysis explored a wider representation of bionetwork in rumen fungi that could be influenced by host-induced factors. However, further study should focus on better characterization of the unclassified fungi in the ruminal microbiome using both cultivation-dependent and cultivation-independent approaches. Additionally, a great deal of work, which is by using the meta-transcriptomic technique, is necessary to elucidate active fungi taxa in the rumen, aiming to gain a better understanding of the rumen fungal community and their dynamic in the rumen.

## Figures and Tables

**Figure 1 animals-10-00144-f001:**
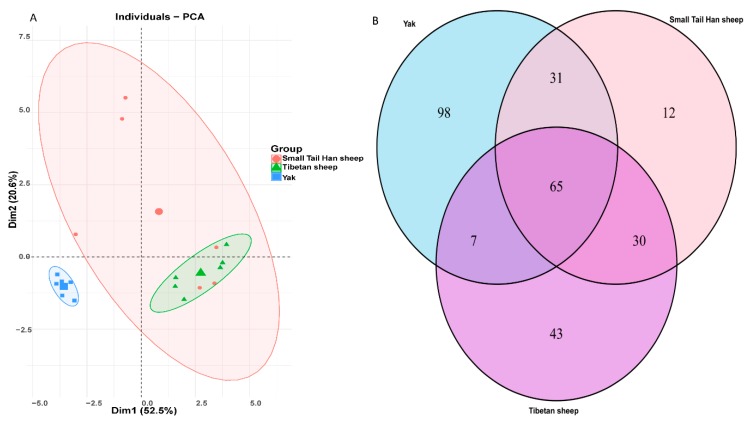
Two-dimensional Principal component analysis (PCA) based on Bray−Curtis distance metrics (**A**); Venn diagrams of OTUs composition (**B**).

**Figure 2 animals-10-00144-f002:**
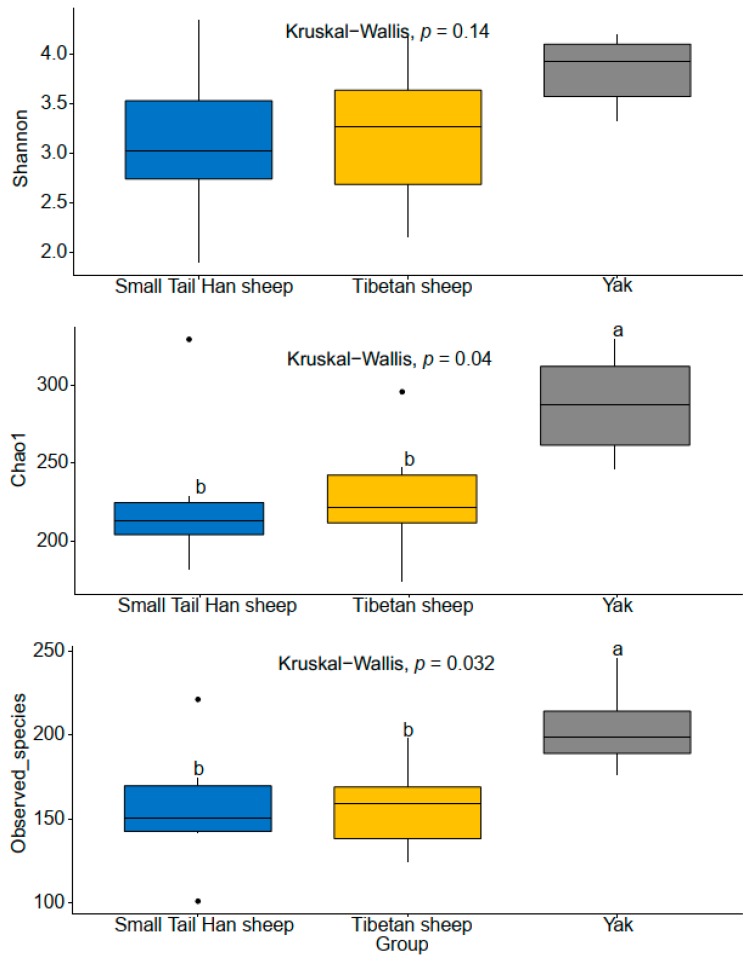
Alpha diversity indices Shannon, Chao1 and Observed_species derived from Tibetan sheep, Small Tail Han sheep and Yak. The Wilcoxon test was used to determine the significant level between two animal groups, while Kruskal−Wallis test was used to calculate the significant level among animal groups, and the *p* value was adjusted using Benjamini−Hochberg method. a, b = different letters indicate a significant difference between animal species (*p* < 0.05).

**Figure 3 animals-10-00144-f003:**
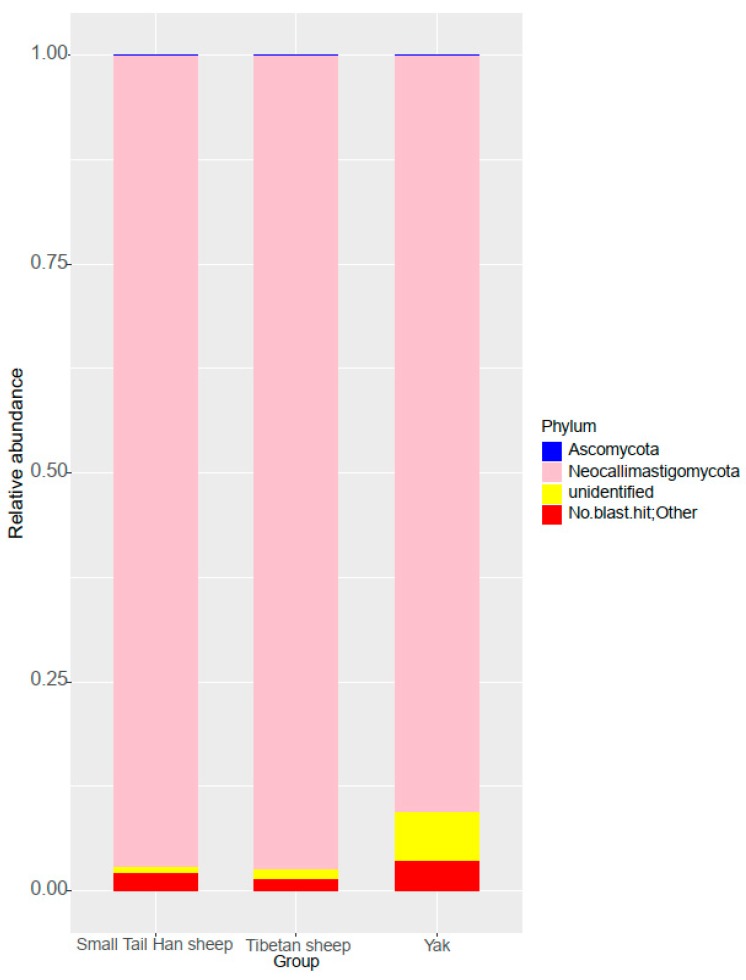
Taxonomic composition of fungal community at the phylum level.

**Figure 4 animals-10-00144-f004:**
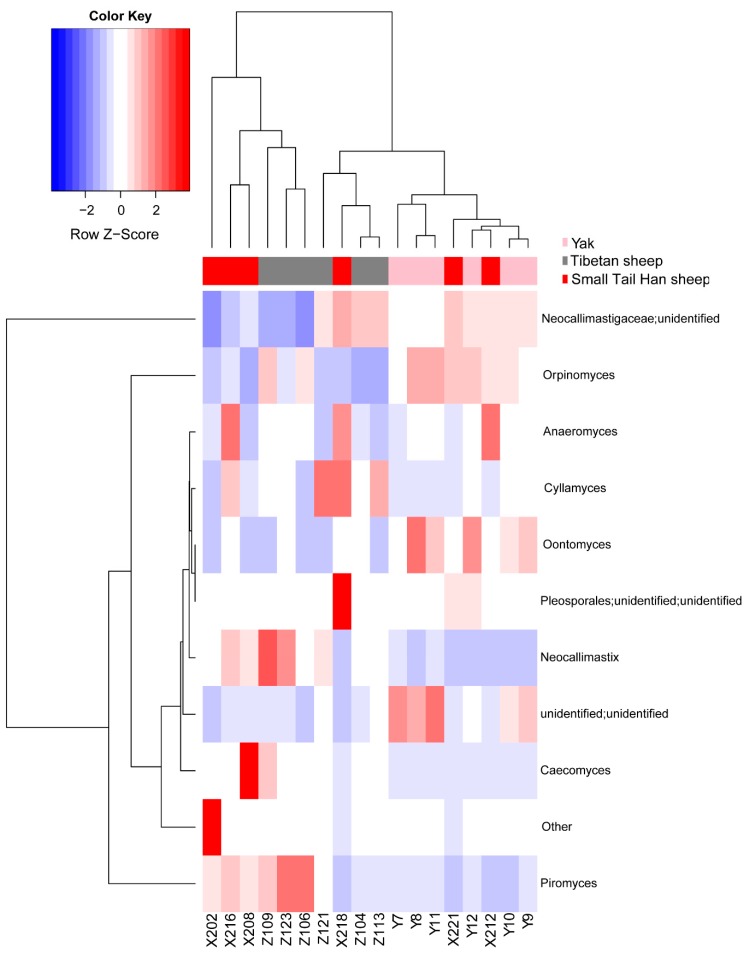
Genus or higher-level fungal profiles of each sample in three animal species, with the values of log10 transformed.

**Figure 5 animals-10-00144-f005:**
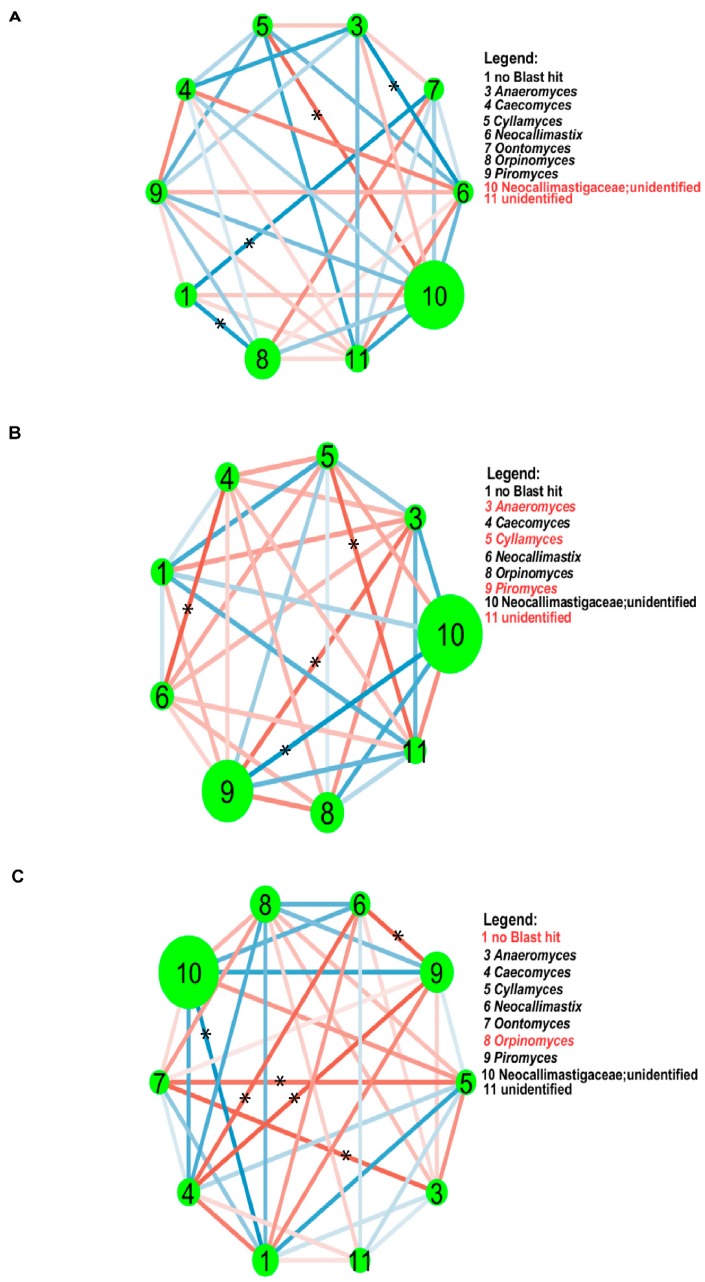
The co-occurrence network analysis of rumen fungi in yak (**A**), Tibetan sheep (**B**) and Small Tail Han sheep (**C**). The size of each node is proportional to the relative abundance. Line in red and line in blue denote positive and negative correlations, respectively. The keystone species were colored by red in the Legend. The correlations between nodes which reached statistically significant levels (*p* < 0.05) were noted with an asterisk (*) in the Figures.

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
