# Peer review of "Characterization of Anaerobic Rumen Fungal Community Composition in Yak, Tibetan Sheep and Small Tail Han Sheep Grazing on the Qinghai-Tibetan Plateau"

_animals, 2020, doi:10.3390/ani10010144_

Round 1
Reviewer 1 Report
Title:
The authors may want to use the term ‘Anaerobic fungal………’ or ‘Anaerobic Rumen Fungal……..’ instead of ‘Rumen fungal………’ as the term rumen fungi is increasingly replaced with anaerobic fungi owing to the detection of these fungi even outside the rumen environments, and even in the non-ruminants.
Simple summary:
Needs to be rephrased. The sentences do not convey the message.
Line 18: associated
Abstract:
Rephrase first sentence
Line 30-31: What does host induced factor means?
Introduction:
The introduction needs to be shortened/ more specific. It looks more like discussion (Line 59-79)
Line 44: Currently 11 genera are known; please refer and cite the latest literature
Line 46: not most of them, all of them belong to order Neocallimasticomycota
Line 66: Rephrase
Line 84: How can we conclude based on culture independent study the fiber degrading ability of detected taxa?
Materials and Methods:
Line 94-95: We know anaerobic fungi remain attached to the fibrous feedstuff. Why did the authors decide to squeeze and use only the liquid part for this study?
Section 2.3: Recent studies have advocated the usage of LSU based primers for proper identification and differentiation of anaerobic fungi. Why such primers amplifying the LSU region were not used?
Results:
Section 3.2: It is interesting to note that all of the identified genera belong to the earlier known genera that were identified solely using ITS region (Prior to 2014). Do you think these results are primer biased? I would like the authors to recheck the presence of newly described members of anaerobic fungal community, especially in the unidentified Neocallimastigaceae members.
Discussion;
Line 242-243: Rephrase
Line 287-288: Cite proper reference here. You may refer to ‘Comparative evaluation of lignocellulolytic activities of filamentous cultures of monocentric and polycentric anaerobic fungi’ to justify your statement.
Reviewer 2 Report
The reviewer has some minor comments before it can be published:
Line 103: Pyro-Seq was not used in this manuscript. Please correct. Line 127: If possible, please only list correlation with p<0.05. Line 148: Please clarify the statistical method in the text or figure legend. Figure 1A is not readable, please enlarge the labeling. Line 178: Please reword this sentence. Figure 4: If possible, please re-plot this heatmap. It will be better if the color can scaled by row rather than by column. According to heatmap, STH sheep clustered together (figure 4), however, high variation was observed in Figure 1. Please add some explanation in the results. The purpose of this study is to find fungal strains that "have the ability to utilize fiber efficiently". According to the data, most of them are un-identified. Please do include some sentences in the discussion, for example, 1) whether the "un-identified" strains is related to efficiency, 2) how to improve the future study to really identify "useful" strains.Author Response
Please see the attachment

Round 2
Reviewer 2 Report
The authors have address all my comments.